# Dynamic aspiration based on Win-Stay-Lose-Learn rule in spatial prisoner's dilemma game

**Zhenyu Shi**[1,2,3,4], **Wei Wei**[1,2,3,4]*, **Xiangnan Feng**[1,2,3,4], **Xing Li**[1,2,3,4], **Zhiming Zheng**[1,2,3,4]

**1** School of Mathematical Sciences, Beihang University, Beijing, China, **2** Key Laboratory of Mathematics, Informatics and Behavioral Semantics, Ministry of Education, Beijing, China, **3** Peng Cheng Laboratory, Shenzhen, Guangdong, China, **4** Beijing Advanced Innovation Center for Big Data and Brain Computing, Beihang University, Beijing, China

* weiw@buaa.edu.cn

**Data Availability Statement:** All relevant data are within the paper and its Supporting information files.

**Funding:** This work is supported by the Research and Development Program of China (No.2018AAA0101100), the Fundamental

## Abstract

Prisoner's dilemma game is the most commonly used model of spatial evolutionary game which is considered as a paradigm to portray competition among selfish individuals. In recent years, Win-Stay-Lose-Learn, a strategy updating rule base on aspiration, has been proved to be an effective model to promote cooperation in spatial prisoner's dilemma game, which leads aspiration to receive lots of attention. In this paper, according to Expected Value Theory and Achievement Motivation Theory, we propose a dynamic aspiration model based on Win-Stay-Lose-Learn rule in which individual's aspiration is inspired by its payoff. It is found that dynamic aspiration has a significant impact on the evolution process, and different initial aspirations lead to different results, which are called *Stable Coexistence under Low Aspiration*, *Dependent Coexistence under Moderate aspiration* and *Defection Explosion under High Aspiration* respectively. Furthermore, a deep analysis is performed on the local structures which cause defectors' re-expansion, the concept of END- and EXP-periods are used to justify the mechanism of network reciprocity in view of time-evolution, typical feature nodes for defectors' re-expansion called Infectors, Infected nodes and High-risk cooperators respectively are found. Compared to fixed aspiration model, dynamic aspiration introduces a more satisfactory explanation on population evolution laws and can promote deeper comprehension for the principle of prisoner's dilemma.

## Introduction

The emergence and stability of cooperative behavior among selfish individuals is a challenging problem in biology, sociology and economics [1]. The prisoner's dilemma(PD) game is considered as a paradigm to portray competition among selfish individuals [2–6]. For general parameter settings, defection is favoured by evolutionary selection, but we can easily observe numerous cooperation phenomenon in various scenarios, e.g., animals will cooperate to obtain food instead of preying alone [7]; companies will set appropriate commodity prices instead of maliciously cutting prices [8]; humans will choose to obey the order instead of jumping in line, etc [9]. Evolutionary game theory provides a practical framework to explain

Research Funds for the Central Universities, the
International Cooperation Project
No.2010DFR00700, Fundamental Research of Civil
Aircraft No. MJ-F-2012-04 and the Beijing Natural
Science Foundation (1192012, Z180005).

**Competing interests:** The authors have declared
that no competing interests exist.

how the cooperation forms [10–14]. Besides, five representative mechanisms considered as
promoting cooperation have been investigated: kin selection, direct and indirect reciprocity,
network reciprocity and group selection [15].

Since the pioneering work of Nowak and May [16], spatial games were proposed and have
attracted ample attention of researchers, in which players are located on the spatially struc-
tured network and only interact with their neighbors. Since then, numerous studies have
emerged to propose various mechanisms which explain the emergence and stability of cooper-
ative behavior, such as punishment [17–20], migration [21–23], game organizers [24, 25],
teaching ability [26–28], and so on. In recent years, aspiration, a parameter representing indi-
vidual's expectation, has attracted many researchers' attention [29–32]. Win-Stay-Lose-Learn
is a representative model based on aspiration, in which one will try to change its strategy only
when its payoff is lower than aspiration [33–35]. Liu and Chen investigated the Win-Stay-
Lose-Learn rules in spatial prisoner's dilemma game [33]; Chu and Liu added the voluntary
participation into the Win-Stay-Lose-Learn rules [34]; Fu studied the stochastic Win-Stay-
Lose-Learn rules in the spatial public goods game [35].

These research held the assumption that an individual player's aspiration is fixed. However,
according to Expected Value Theory and Achievement Motivation Theory proposed by Atkin-
son [36], one's aspiration will be influenced by its previous payoff, and individuals tend to
lower their aspirations in interactions in a crisis [37]. If one's payoff is higher than its aspira-
tion, the aspiration tends to be increased, otherwise decreased, some researchers also paid
attention to this and had some related researches. [37–39]. In this paper, a dynamic aspiration
model is introduced based on Win-Stay-Lose-Learn rules, and the principle of defection's
expansion or cooperation's survival under the dynamic aspiration model is investigated. The
rest of our paper is organized as follows. First the detailed model of the dynamic aspiration
based on Win-Stay-Lose-Learn rules is shown. Then the main results under our model is pro-
vided by four parts: *Overview*, *Stable Coexistence under Low Aspiration*, *Dependent Coexistence
under Moderate aspiration* and *Defection Explosion under High Aspiration*. The concept of
enduring (END) and expanding (EXP) periods [40–42] are also used to justify the mechanism
of network reciprocity in view of time evolution and find typical feature nodes called Infectors,
Infected nodes and High-risk cooperators respectively. Finally the wider implications of our
work and the direction of the future research are discussed.

## Model

Our model is described as follows. We use the $L \times L$ square lattice with periodic boundary con-
ditions. Each node represents a player who has one of the following two strategies: cooperation
($\mathcal{C}$), or defection($\mathcal{D}$). The strategy of node $i$ is represented as $s_i$, and node $i$'s aspiration is repre-
sented as $A_i$. At the beginning of the evolutionary process, each node is given [an] initial $s_i$ and
$A_i$. The evolutionary process is performed step by step until the network is stable. In each step,
players synchronously update their strategies and aspirations as follows:

(a)**Rule of game:** Each node $i$ plays the prison's dilemma game with its four neighbors and
gets the payoff $P_i = \sum_{j \in \Omega_i} P_{ij}$, where $\Omega_i$ denotes the neighbors of node $i$. $P_{ij}$ is $i$'s payoff when $i$

plays the game with $j$ and it is got by Table 1, and if both of their strategies are $\mathcal{C}$ or $\mathcal{D}$, they
will get the reward $R$ or punishment $P$. If $i$'s strategy is $\mathcal{C}$ and $j$'s strategy is $\mathcal{D}$, $i$ will get the
sucker's payoff $S$ and the $j$ will get the temptation value $T$, vice versa. In prison's dilemma
game, the above parameters meet $T > R > P > S$.

**Table 1. Payoff matrix of prison's dilemma game.**

|  | $\mathcal{C}$ | $\mathcal{D}$ |
|---|---|---|
| $\mathcal{C}$ | $R$ | $S$ |
| $\mathcal{D}$ | $T$ | $P$ |

Without loss of generality, we set $R = 1$ and $P = 0$. And to ensure single parameter, there are some typical representative sub-classes of PD game, e.g., Donor & Recipient (D & R) game which assumes $T + S = 1$ [43–49] and boundary game which assumes $t = b$ and $S = 0$ [16, 33]. In this paper, the boundary game is used as what we mainly study is the impact of $T$ on evolution process. Thought it has $S < 0$ in PD games, our experiment shows that the Monte Carlo simulation result is almost the same as $S = 0$ when $S$ is close to 0(for instance, $S = -0.01$), so it is assumed that $S = 0$ in boundary games.

(b) **Rule of strategy's update:** Each node $i$ chooses one of its four neighbors $j$ randomly with equal probability. If $i$'s payoff is lower than its aspiration $A_i$, $i$ will be *dissatisfied* and choose to adopt $j$'s strategy with the probability:

$$W_{ij} = \frac{1}{1 + \exp\left[(P_i - P_j)/K\right]},\tag{1}$$

where $K$ stands for the amplitude of noise [6]. Without loss of generality, we use $K = 0.1$ in our model [50, 51]. If $i$'s payoff is higher than or equal to its aspiration, $i$ will be *satisfied* and not change its strategy.

(c) **Rule of aspiration's update:** Each node $i$ updates its aspiration by the formula:

$$A_i(t + 1) = A_i(t) + a * (P_i(t) - A_i(t)),\tag{2}$$

where $A_i(0) = A$ is given for all the nodes initially. Based on Achievement Motivation Theory [36], in a representative model, one's aspiration is changed concerning its payoff linearly. So we introduce the dynamic aspiration by the parameter $a$, where $a \in [0, 1]$ stands for the sensitivity of aspiration. Higher $a$ means individual's aspiration is changed more drastically by its payoff. Previous work with fixed aspiration is equivalent to the case $a = 0$, and $a = 1$ means that individual's aspiration totally depends on its payoff from the last step. Considering that one's aspiration should not be changed too drastically, we set $a = 0.01$ in our model.

The step (a)-(c) will repeat 100,000 times in one simulation. The fraction of cooperators and defectors at step $t$ are denoted as $r_{\mathcal{C}t}$ and $r_{\mathcal{D}t}$ respectively. And for each parameter, we perform 20 independent simulations to get $r_{\mathcal{C}}$, the fraction of cooperators when stable, which is thought as the main index to measure a network's cooperation level.

## Results

### 0.1 Overview

Our experiment is performed on the $100 \times 100$ square lattice with periodic boundary conditions. In initial, cooperators and defectors are distributed uniformly at random occupying half of the square lattice respectively, and all the nodes are given the same initial aspiration $A$. As the main parameters, we consider the initial aspiration level $A$ and the temptation to defect $b$. Fig 1 presents the fraction $r_{\mathcal{C}}$ of cooperators when stable as a function of the $b$ for different aspiration levels. It is found that the aspiration level has a significant influence on $r_{\mathcal{C}}$. Three

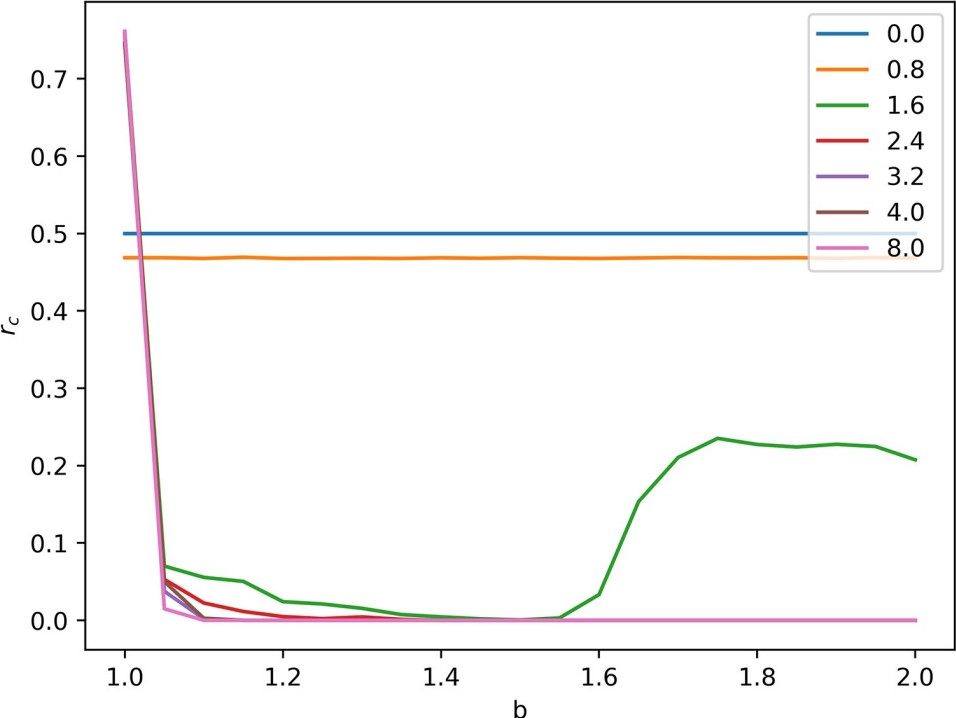

**Fig 1. Average fractions of cooperation when stable as a function of *b* for different values of the initial aspiration *A*, as obtained by means of simulations on square lattices.**

different phases, *Stable Coexistence under Low Aspiration*, *Dependent Coexistence under Moderate aspiration* and *Defection Explosion under High Aspiration* could be observed for different values of *A*. For small values of *A* (*A* = 0 and *A* = 0.8), $r_c$ is equal/close to 0.5 no matter what value of *b* is. For large values of *A* (*A* ≥ 2.4), cooperators could hardly survive for any *b* > 1.0. An interesting phenomenon is discovered for moderate values of *A* (*A* = 1.6) that when *b* is lower than 1.6, cooperators could not survive and $r_c$ almost equals 0, but when *b* is higher than 1.6, cooperation could survive and $r_c$ is bigger than 0.2. According to common sense, the higher *b* is, the harder cooperators survive, which is inconsistent with our experimental results. Above unusual phenomenon is the focus in our paper.

## 0.2 Stable Coexistence under low aspiration (*A* ≤ 1.0)

For small values of *A*, individual's aspiration is easy to be satisfied so cooperators and defectors can coexist. For *A* = 0, all the nodes are satisfied and never change their strategies, so $r_c$ will always keep 0.5. And for *A* = 0.8, there are only a few nodes whose four neighbors are all defectors will be dissatisfied: if it is a cooperator, it will change its strategy to $\mathcal{D}$; if it is a defector, it will be always dissatisfied but can't change its strategy since all his neighbors' strategies are the same. At the beginning, $r_0 = r_{\mathcal{D}0} = 0.5$, so the frequency of cooperators with four neighbors being all defectors could be calculated approximately:

$$r = r_{C0} * r_{\mathcal{D}0}^4 = \frac{1}{32} \approx 0.03, \tag{3}$$

which is consistent with the result $r_c = 0.47$ in the simulation experiment and $r_c \approx r_{c0} - r$. In fact, the above result holds for all *A* ∈ (0, 1], where the proportion of cooperators dissatisfied is

only $r_{\mathcal{D}0}^A$ and other cooperators will always be satisfied and survive. To conclude, for small values of $A$, initial cooperators could coexist with defectors and neither of them could expand. Low aspiration means both strategies and aspirations are long-term stable. This case is called *Stable Coexistence under Low Aspiration*.

## 0.3 Dependent Coexistence under moderate aspiration ($1.0 < A \leq 2.0$)

For moderate values of $A$, cooperators can't survive for small values of $b$. Fig 2 shows the spatial distributions of strategies and aspirations at different time steps $t$ for $A = 1.6$ and $b = 1.2$. The evolution process can be divided into the following stages:

- At first, every node with strictly less than two $\mathcal{C}$ neighbors is dissatisfied. During this phase, defectors expand quickly and cooperators try to endure defectors' invasion, which is the so-called *END period* [40–42]. $r_{\mathcal{C}}$ decreases and the aspirations of all the nodes are lower than 2.0.

- When $t = 10$, some cooperators still survive by forming some clusters in the END period. The defectors neighboring with the clusters are dissatisfied and have lower payoffs than their $\mathcal{C}$ neighbors, so these clusters could expand by converting the neighboring defectors to cooperators, which is called the *EXP period* [40–42]. Dissatisfied defectors may evolve into cooperators gradually. Meanwhile, the aspirations of these cooperators gradually rise up.

- When $t = 100$, cooperators have expanded fully during the EXP period and $r_{\mathcal{C}}$ reaches a maximum. However, during the process of cooperation's expansion, cooperations' aspirations rise up gradually. The aspirations of the nodes on the boundary of the cooperators' clusters are higher than 2.0 now. These cooperators are no longer satisfied if they neighbor with two defectors so they will re-evolve into defectors. This leads to a repeated evolution process on the boundaries of the cooperators' clusters where nodes evolve into cooperators and defectors repeatedly. At the same time, nodes inside the cooperators' clusters are still satisfied so they keep their strategies and their aspirations rise up gradually.

- When $t = 200$, defectors gradually penetrated into the cooperators' clusters. These cooperators' aspirations are now close to 4.0 so once they neighbor with a defector, they will be dissatisfied and also change into defectors gradually. As a result, chain phenomenon happens that defectors almost occupy the entire network and the cooperators almost disappear rapidly.

One can see that although cooperators can survive in the END period and expand in the EXP period by forming clusters, defectors finally occupy the network when it is stable. The network reciprocity is undermined by dynamic aspirations. In dynamic aspiration models, cooperators' aspirations will become too high to endure defectors' re-expansion because of the long-term satisfaction. which is different from the fixed aspiration model. Fig 3 shows the probability that cooperators could survive as a function of the cooperators' initial proportion $r_{C0}$ for $A = 1.6$ and $b = 1.2$. For every different $r_{C0}$, 100 independent experiments is performed. Under these parameters, cooperators are possible to survive only when $r_{C0} > 0.95$, and sure to survive only when $r_{C0} > 0.99$. In other words, defectors could expand and occupy the network finally even if they are very few initially. This result is different from the result of fixed aspiration model and it may be caused by some special local structures which could lead to the high aspirations of cooperators.

Fig 4 shows all possible local structures in the network for $A = 1.6$ and $b = 1.2$. When a node has two or less $\mathcal{D}$ neighbors, it is satisfied and won't change its strategy and its aspiration will

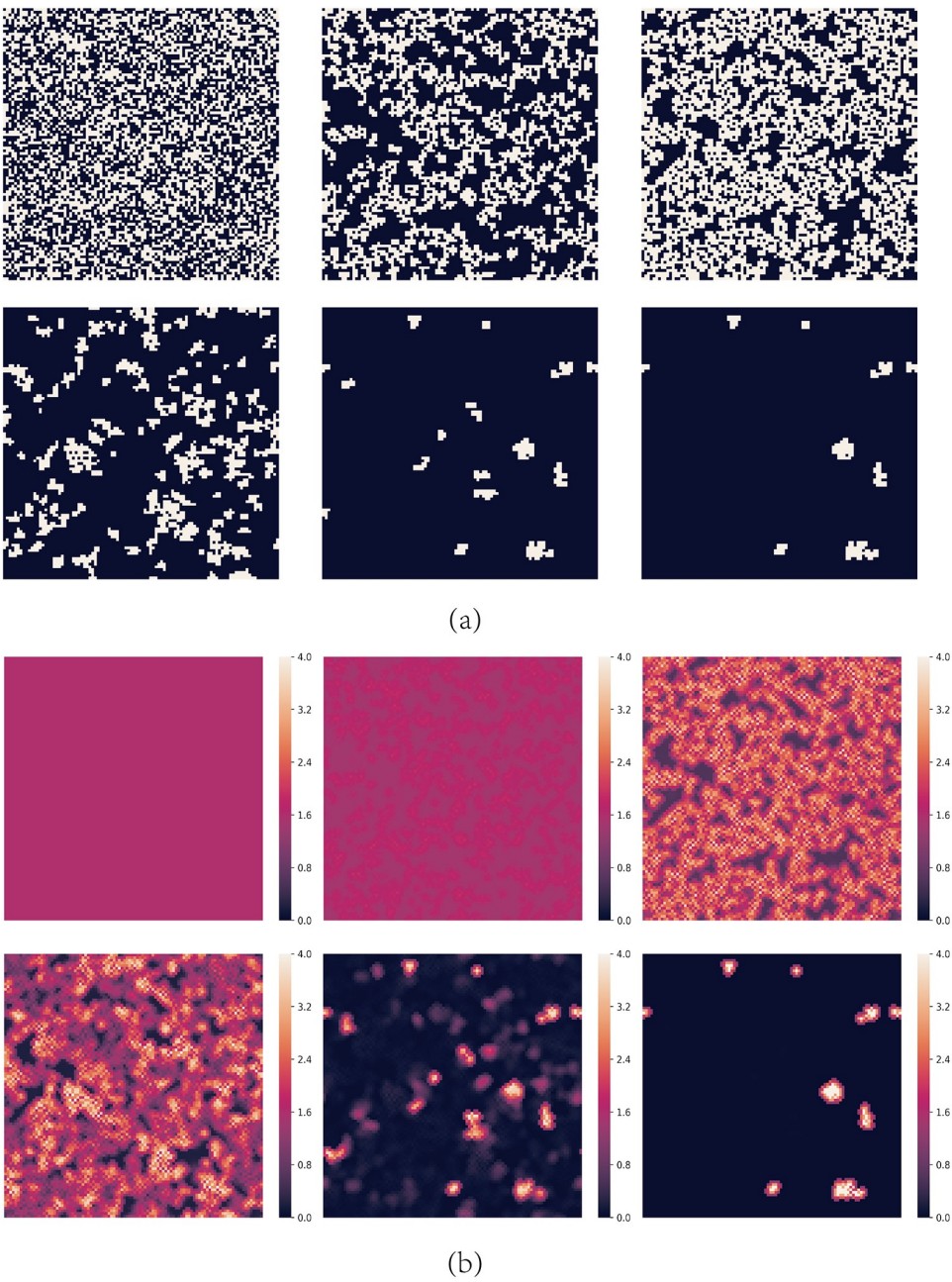

**Fig 2. Snapshots of typical distributions of strategies and aspirations at different time steps $t$ for $A = 1.6$ and $b = 1.2$.** (a) represents strategies, where cooperators are depicted white and defectors are depicted black. (b) represents aspirations. The steps of them are $t = 0, 10, 100, 200, 500$ and $1000$ respectively.

keep lower than its payoff. When a node has four $\mathcal{D}$ neighbors, it is always dissatisfied. But since all its neighbors are defectors, it can only be a defector by copying neighbors' strategies and won't change its strategy any more, and its aspiration will keep decreasing.

Now we consider the structure that a node has three $\mathcal{D}$ neighbors and one $\mathcal{C}$ neighbor. Fig 5 shows the detailed principle for defectors' expanding under this initial structure. The evolution process can be divided into the following stages:

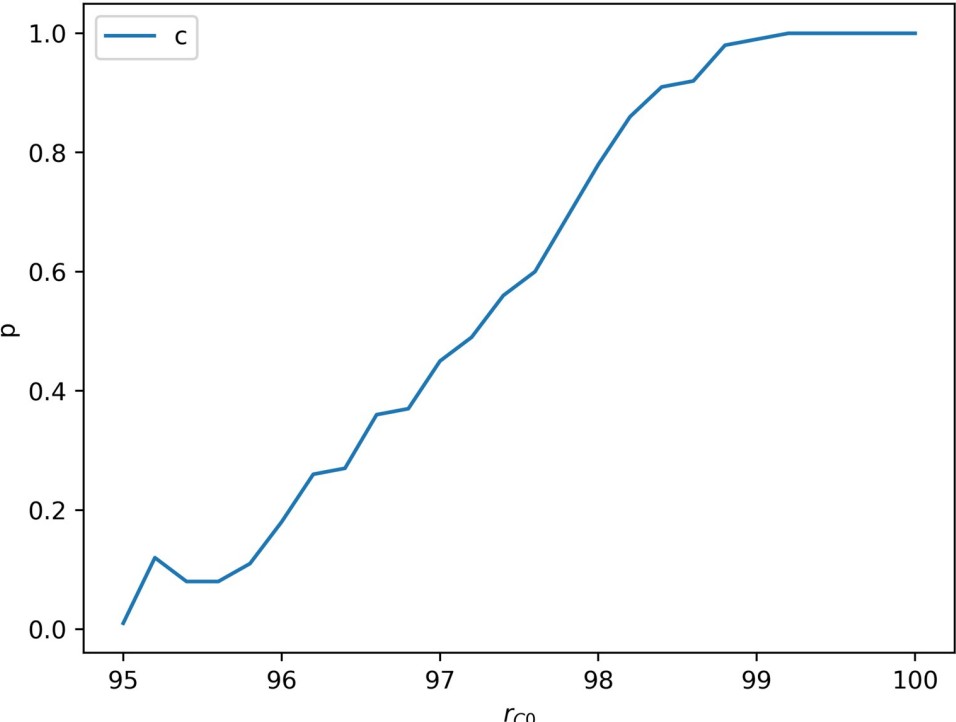

**Fig 3. The probability that cooperators can survive as a function of the cooperators' initial proportion $r_{C0}$ for $A = 1.6$ and $b = 1.2$.** Cooperators are easier to survive when $r_{C0}$ is higher. They are possible to survive only when $r_{C0} > 0.95$, and sure to survive only when $r_{C0} > 0.99$.

- When $t = 0$, the only one node dissatisfied is node $X$ because its aspiration is 1.0. Since it has three D neighbors and one $\mathcal{C}$ neighbor, $X$ will evolve into cooperator and defector repeatedly. We call such node as an *Infector*. As a result, node $Y$'s payoff is sometimes 4.8 and sometimes 3.6, which we called an *Infected node*, and $Y$'s aspiration at step $t$ can be got by the recursive equation:

$$A_Y(t) = \begin{cases} A_Y(t-1) + a * (4.8 - A_Y(t-1)), s_Y = \mathcal{D}, \\ A_Y(t-1) + a * (3.6 - A_Y(t-1)), s_Y = \mathcal{C}. \end{cases} \tag{4}$$

- With $t$ growing, we can easily prove that $A_Y$ will be higher than 3.6. Next time when $X$ evolves into a defector, $P_Y = 3.6 < A_Y$, so $Y$ is dissatisfied. $Y$ has three $\mathcal{C}$ neighbors, so it will be easy to evolve into a cooperator. But when $Y$ evolves into cooperator, its payoff will decrease and it is still dissatisfied, so $Y$ will also evolve into cooperator and defector repeatedly. In other words, the Infected node $Y$ also becomes an Infector. For the same reason, node $Z$'s payoff is sometimes 4.0 and sometimes 3.0, $Z$ becomes an Infected node and $Z$'s aspiration at step $t$ can be got by the recursive equation:

$$A_Z(t) = \begin{cases} A_Z(t-1) + a * (4.0 - A_Z(t-1)), s_Z = \mathcal{D}, \\ A_Z(t-1) + a * (3.0 - A_Z(t-1)), s_Z = \mathcal{C}. \end{cases} \tag{5}$$

- With $t$ further growing, we can easily prove that $A_Z$ will be higher than 3.0. Next time when $Y$ evolves into a defector, $P_Z = 3.0 < A_Z$, so $Z$ is dissatisfied and may evolve into a defector in a few steps. Now $Z$'s other neighbors' aspirations are all near to 4.0, which we call *High-risk*

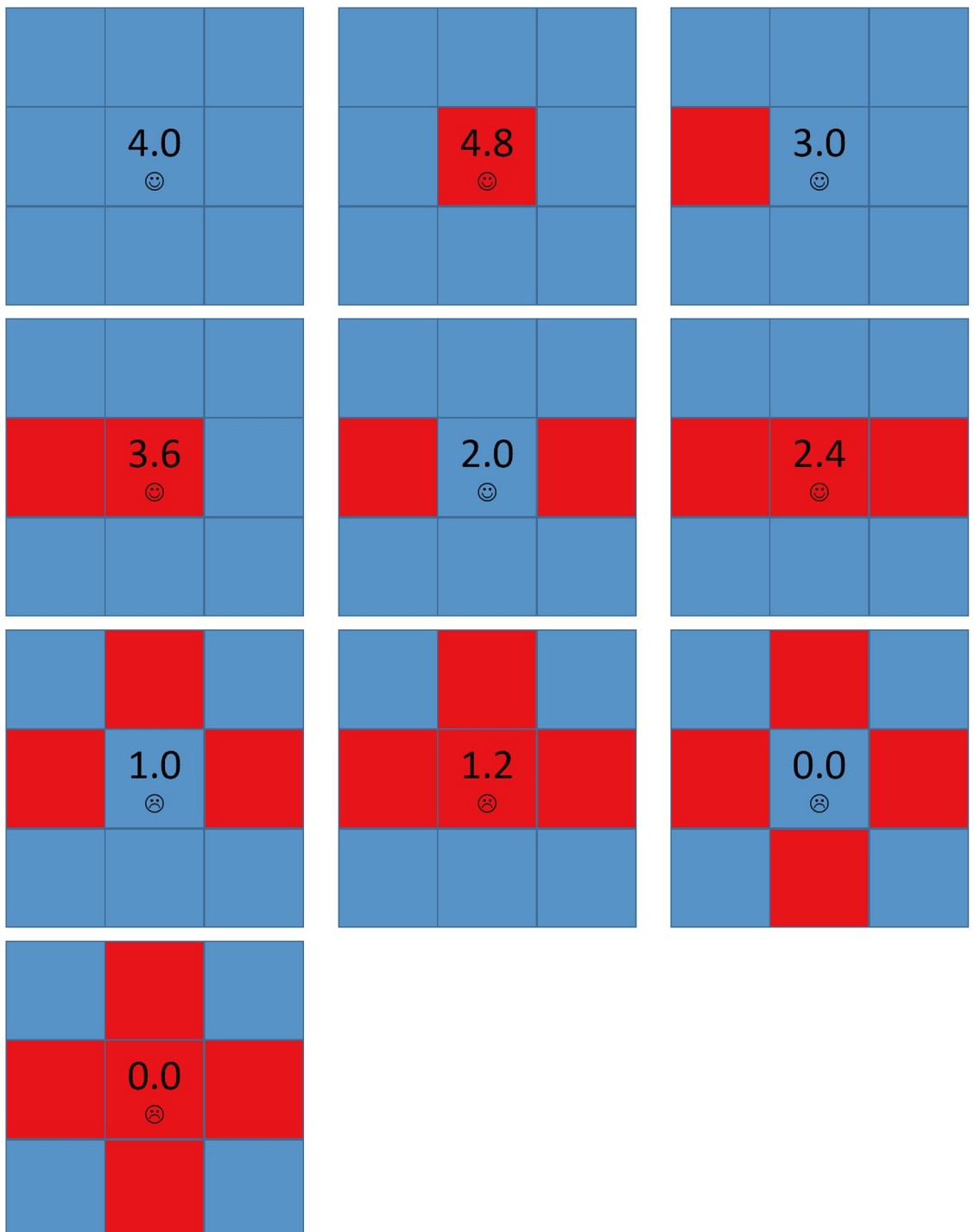

**Fig 4. The local structures of strategies for $A$ = 1.6 and $b$ = 1.2.** Each square corresponds to a single player, where cooperators are depicted blue and defectors are depicted red. Value denoted in the center square is the individual's payoff. Smiling face represents satisfaction while crying face represents dissatisfaction.

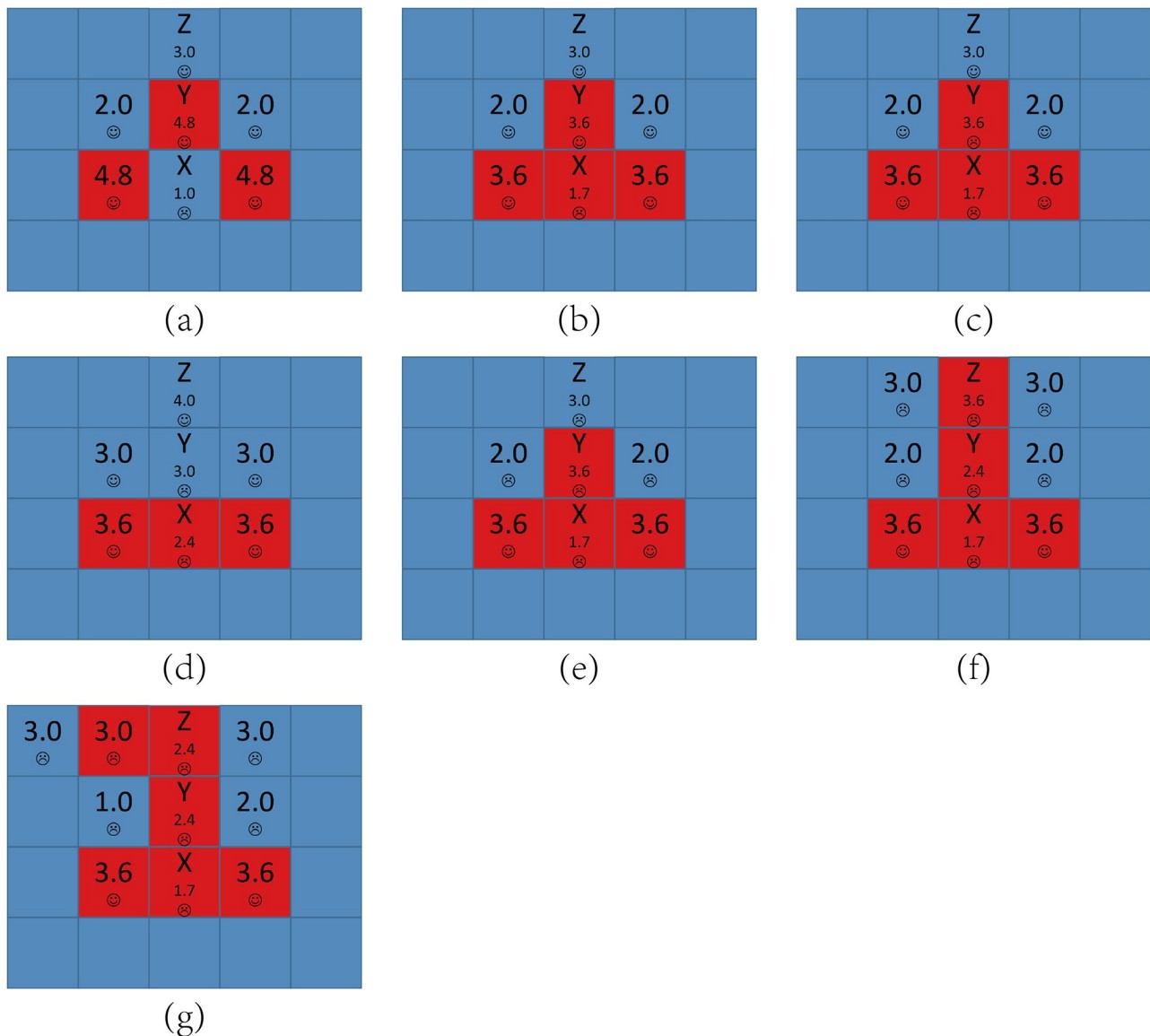

**Fig 5. The detailed principle for defectors' expanding for *A* = 1.6 and *b* = 1.2.** A node is surrounded by three defectors and one cooperator initially. Smiling face represents satisfaction while crying face represents dissatisfaction.

*cooperator*. Once *Z* evolves into a defector, their payoffs decrease to 3.0 so they are dissatisfied and may evolve into defectors, too.

- Furthermore, almost every node's aspiration in the network is near to 4.0 because their payoffs have been keeping 4.0 for a long time. In other word, all the cooperators in the network have became High-risk cooperators. As a result, for each cooperator *i*, once one of *i*'s neighbors evolves into a defector, *i* may evolve into a defector soon, which is a chain phenomenon and causes defectors' expanding.

Fig 6 shows the spatial distributions of strategies and aspirations at different time steps *t* for *A* = 1.6, *b* = 1.2 with the above initial structure, from which we can also get the expansion trajectory by the aspiration distribution.

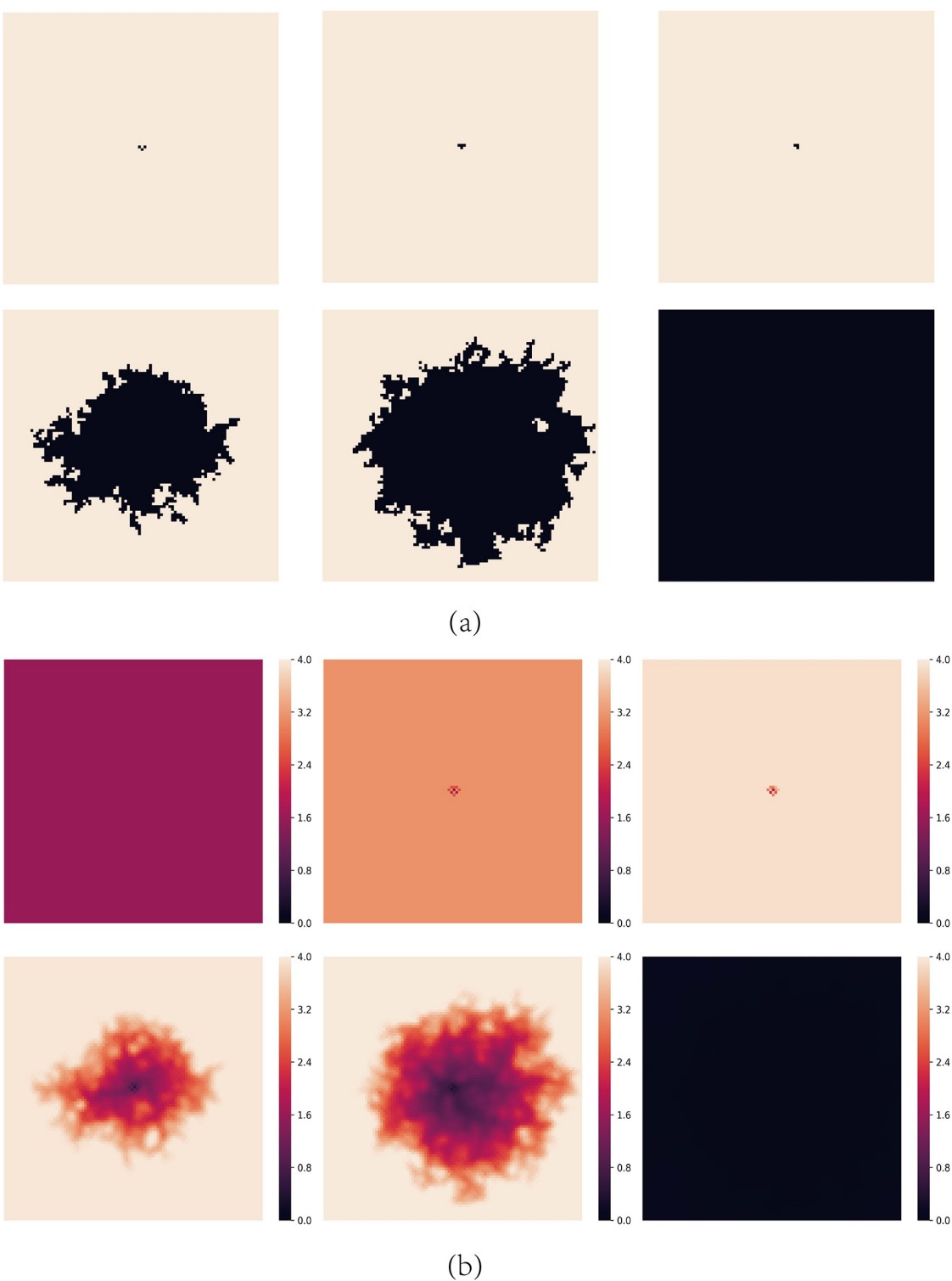

**Fig 6. Snapshots of typical distributions of strategies and aspirations at different time steps *t* under the initial structure shown in Fig 4 for *A* = 1.6 and *b* = 1.2.** (a) represents strategies, where cooperators are depicted white and defectors are depicted black. (b) represents aspirations. The steps of them are *t* = 0, 10, 100, 200, 500 and 1000 respectively.

In the network with random setup, during the END and EXP period, cooperators will survive and expand by the mechanism of network reciprocity. But once there is at least one Infector who has three $\mathcal{D}$ neighbors and one $\mathcal{C}$ neighbor initially, defectors will re-expand to the whole network. If $r_{\mathcal{D}0} > 0.05$, such an Infector almost certainly exists, so defectors almost certainly expand to the whole network. Cooperators can survive only when there is no Infector initially in the network.

However, for large values of $b$, cooperators can partially survive. Fig 7 shows all possible local structures in the network for $A = 1.6$ and $b = 1.7$. Compared to Fig 4, if a cooperator has three $\mathcal{D}$ neighbors and one $\mathcal{C}$ neighbor, it will be dissatisfied and may evolve into a defector. But once it evolves into a defector, it becomes satisfied and doesn't change any longer, and the network becomes stable. In other word, there is no Infector existing in the network. Fig 8 shows the spatial distributions of strategies and aspirations at different time steps $t$ for $A = 1.6$, $b = 1.7$ with the above initial structure. Cooperators can survive by forming clusters, but as the values of $b$ is higher, these clusters couldn't expand. The above evolution process only goes through the END period and then the network has been stable.

From the above we know the main difference between $b < 1.6$ and $b \geq 1.6$ for $A = 1.6$ is whether a defector who has three $\mathcal{D}$ neighbors and one $\mathcal{C}$ neighbor is satisfied, more simply, whether an Infector exists. For $b < 1.6$, $P = b < A$, the defector is dissatisfied and will evolve into cooperator and defector repeatedly and it is an Infector, which causes the chain phenomenon leading to defectors' expansion shown in Fig 5. But for $b \geq 1.6$, $P = b \geq A$, the defector is satisfied and the network will be stable soon so that cooperators can survive in the end. In fact, for all the $A \leq 2.0$, the conclusion is the same and phase transition happens in $b = A$ because $A$ is the highest values of $b$ which allows Infectors to exist.

To conclude, for moderate values of $A$, cooperators will survive and expand in the early stages of evolution when $b$ is lower than $A$, which are END and EXP periods respectively. But according to our results, the existence of Infectors may lead to defectors' re-expansion. The core reason for this phenomenon is that the cooperators increase their aspirations excessively and become the so-called High-risk cooperators, which needs to be vigilant in the evolution process of cooperation.

## 0.4 Defection Explosion under High Aspiration ($A > 2.0$)

For $A = 2.4$, cooperators can survive only when $r_{\mathcal{C}0}$ is high enough for any $b > 1$, on the contrary, only a few defectors initially can lead cooperators to die out. Fig 9 shows all possible local structures in the network for $A = 2.4$ and $b = 1.7$. If a cooperator has two $\mathcal{D}$ neighbors, it is dissatisfied and evolves into a defector soon, and then it is satisfied. If a cooperator or defector $X$ has three $\mathcal{D}$ neighbors, it is dissatisfied and evolves into cooperator and defector repeatedly. $X$'s payoff is 1.0 when it is a cooperator and 1.7 when it is a defector. $X$'s aspiration at step $t$ can be got by the recursive equation:

$$A_X(t) = \begin{cases} A_X(t-1) + a*(1.7 - A_X(t-1)), s_Y = \mathcal{D}, \\ A_X(t-1) + a*(1.0 - A_X(t-1)), s_Y = \mathcal{C}. \end{cases} \tag{6}$$

With $t$ growing, we can easily prove that $A_X$ will be lower than 1.7. And next time when $X$ evolves into a defector, it will be satisfied and don't change its strategy any more. If a cooperator or defector has four $\mathcal{D}$ neighbors, it will be never satisfied, but it can only be a defector since all its neighbors are defectors. All the simple local structures can't lead defectors to expand.

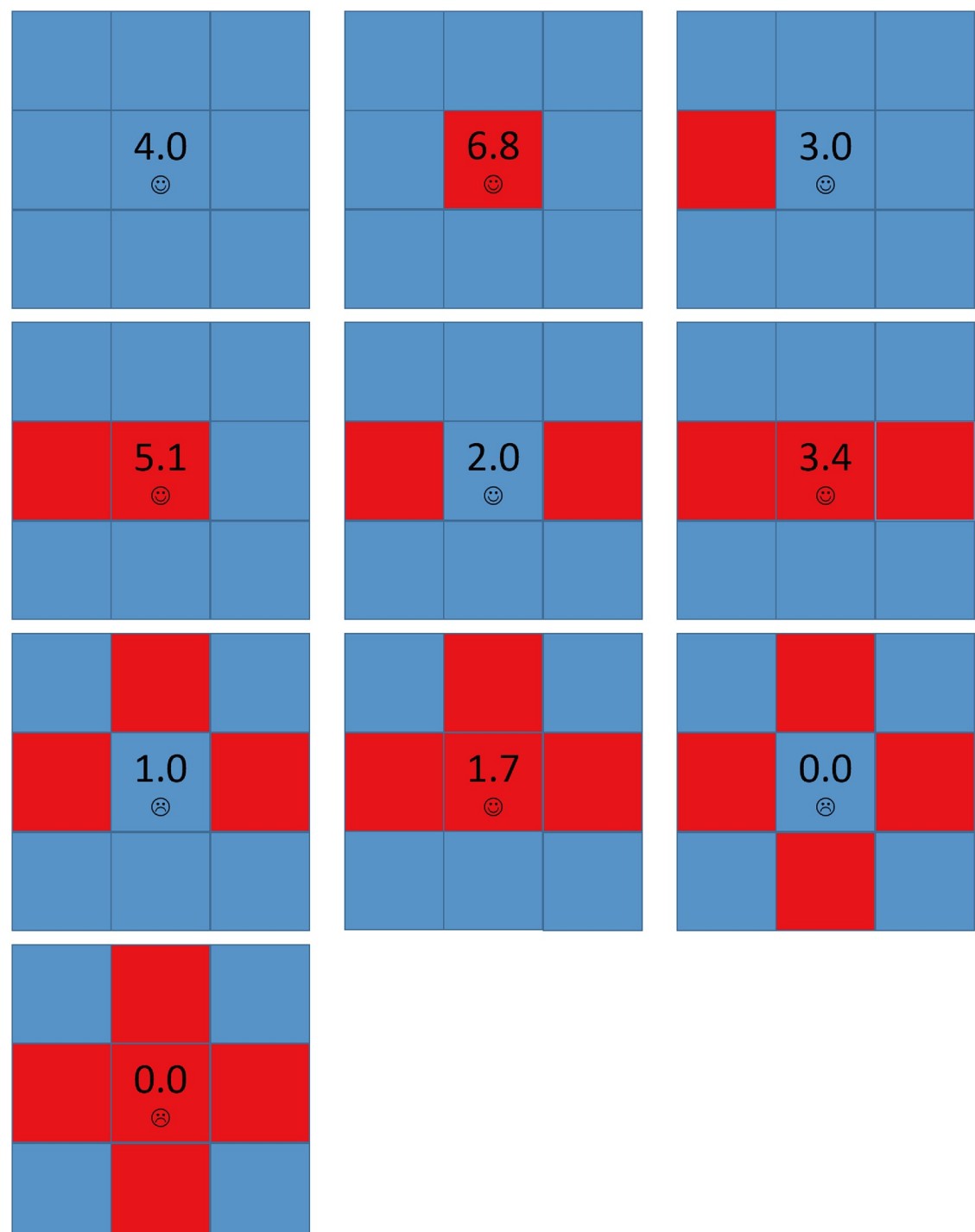

**Fig 7. The local structures of strategies for *A* = 1.6 and *b* = 1.7.** Each square corresponds to a single player, where cooperators are depicted blue and defectors are depicted red. Value denoted in the center square is the individual's payoff. Smiling face represents satisfaction while crying face represents dissatisfaction.

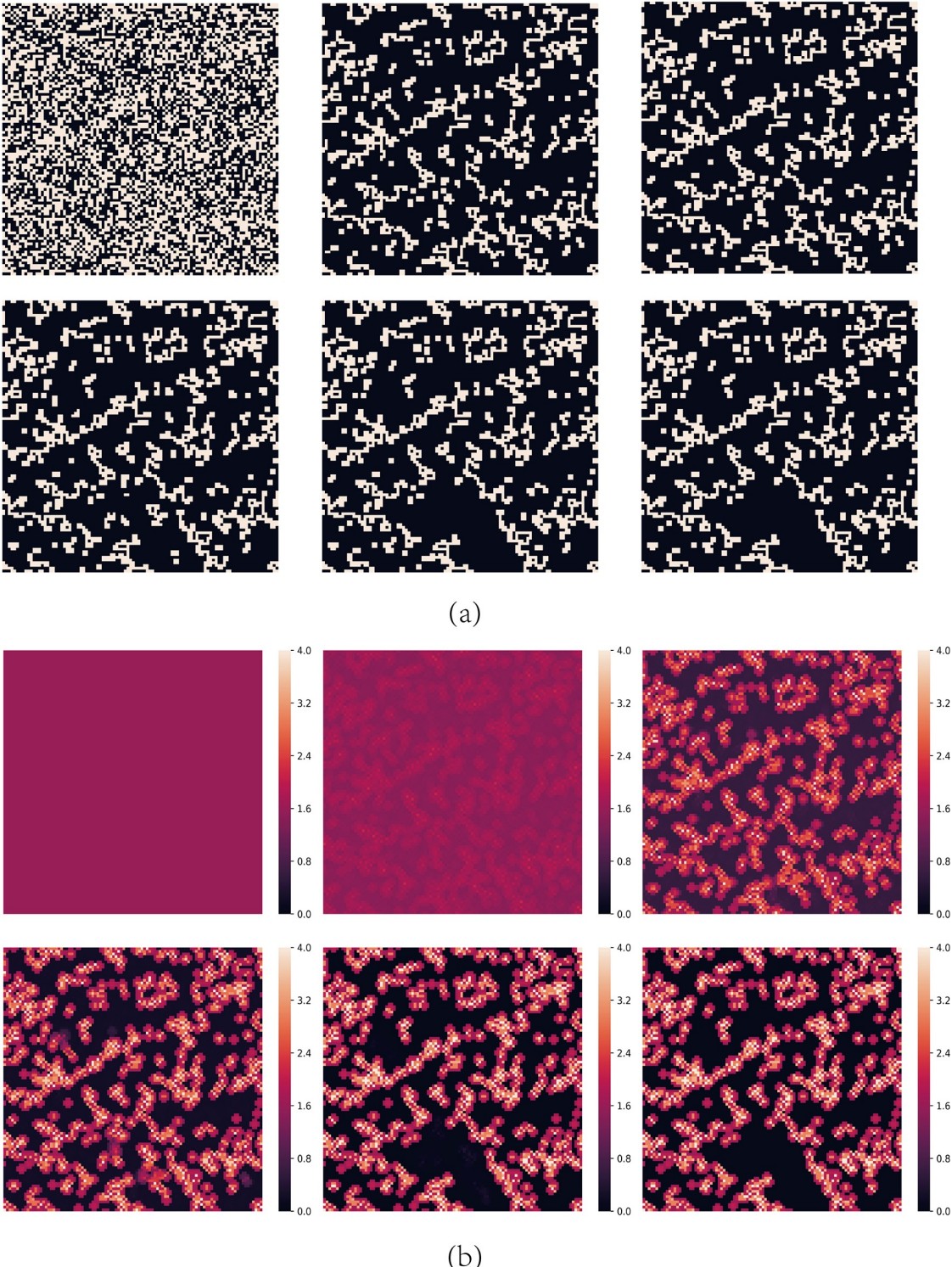

**Fig 8. Snapshots of typical distributions of strategies and aspirations at different time steps $t$ for $A$ = 1.6 and $b$ = 1.7.** (a) represents strategies, where cooperators are depicted white and defectors are depicted black. (b) represents aspirations. The steps of them are $t$ = 0, 10, 100, 200, 500 and 1000 respectively.

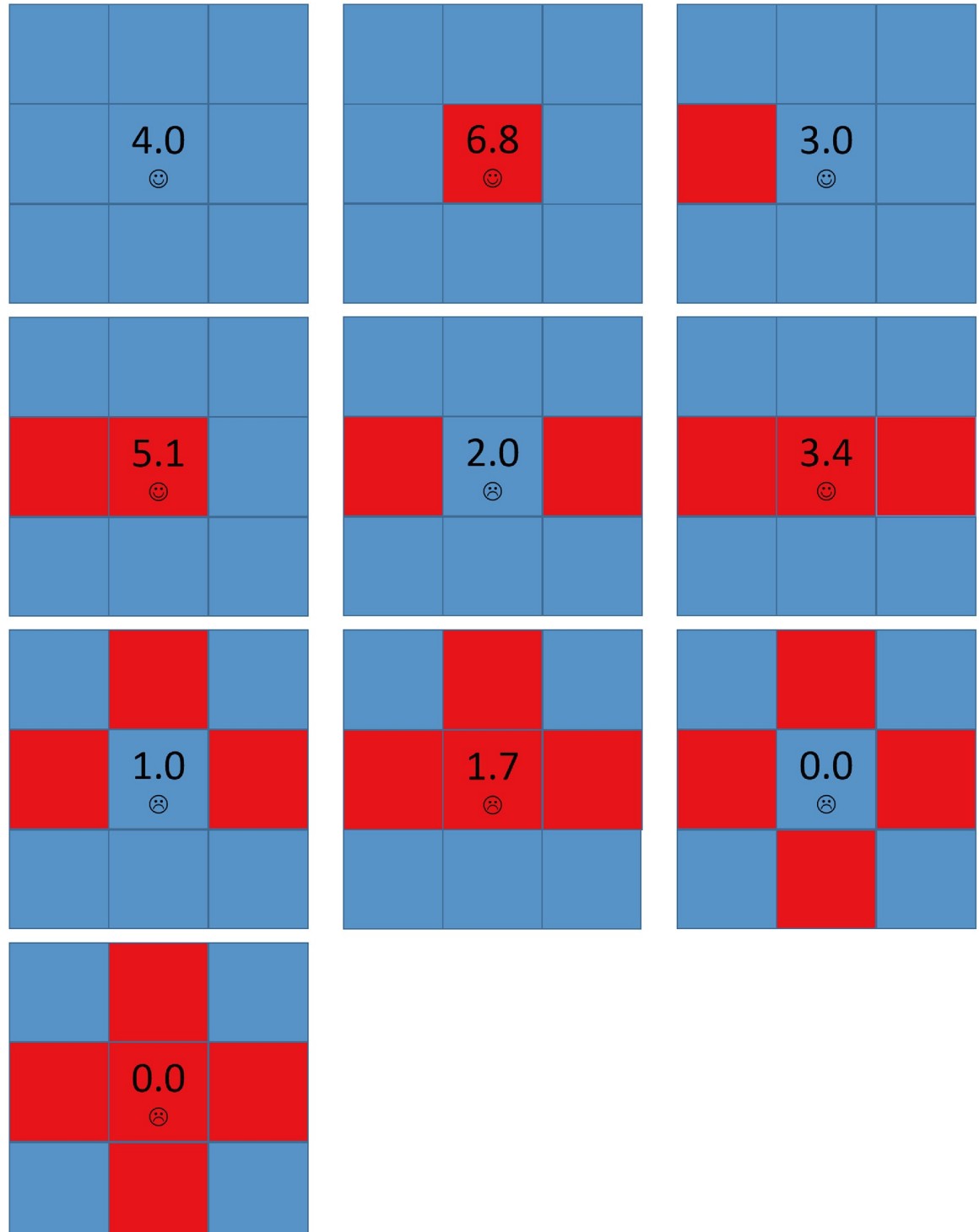

**Fig 9. The local structures of strategies for *A* = 2.4 and *b* = 1.7.** Each square corresponds to a single player, where cooperators are depicted blue and defectors are depicted red. Value denoted in the center square is the individual's payoff. Smiling face represents satisfaction while crying face represents dissatisfaction.

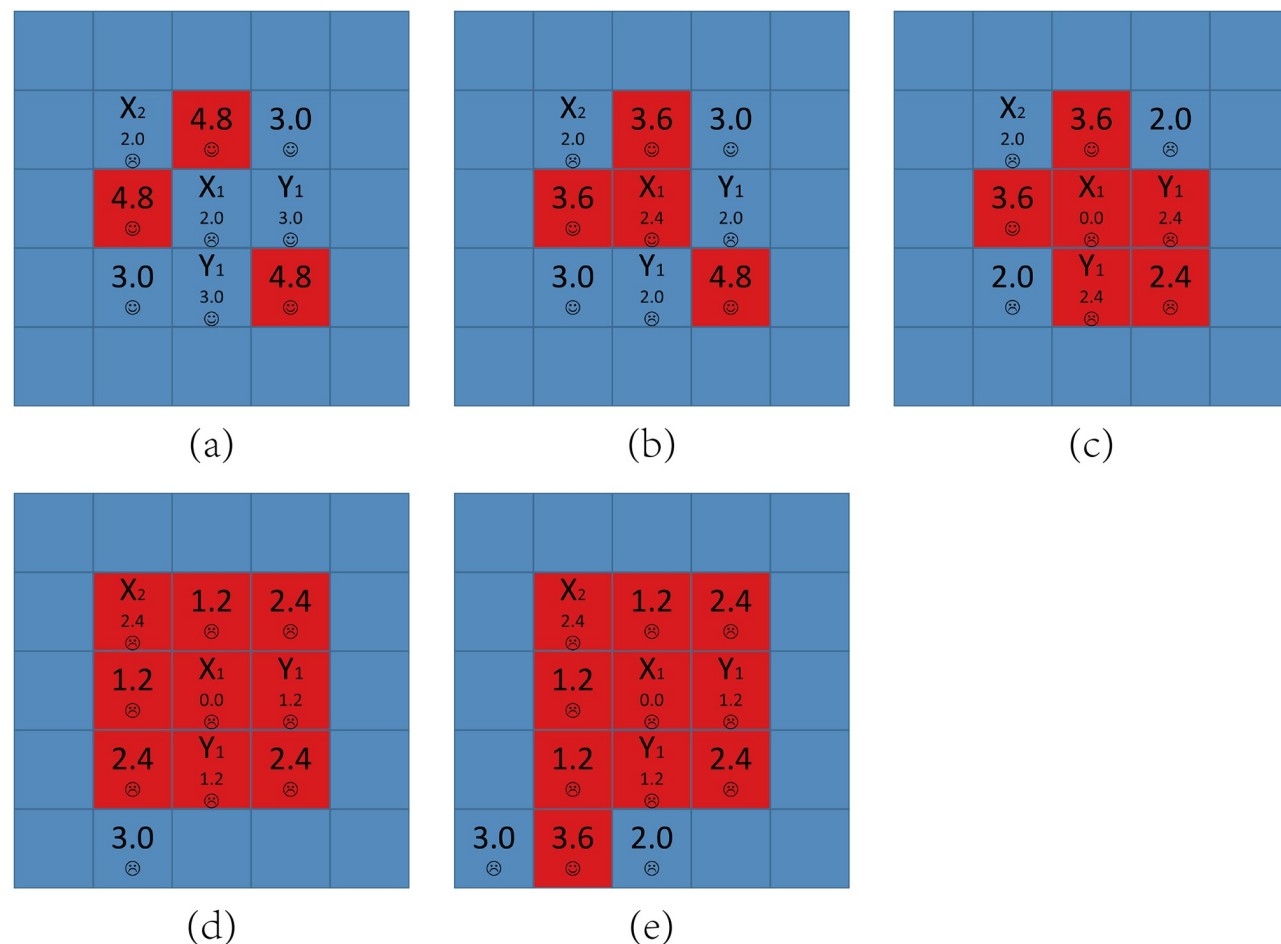

**Fig 10. The detailed principle for defectors' expanding for $A = 2.4$, $b = 1.7$.** The initial local structure is shown in (a). Smiling face represents satisfaction while crying face represents dissatisfaction.

Fig 10 shows the structure which causes the defectors' expansion. In initial, nodes $X_1$ and $X_2$ are dissatisfied and may evolve into defectors. Once $X_1$ evolves into a defector, nodes $Y_1$ and $Y_2$ become dissatisfied and may also evolve into defectors, so do the other five nodes. All the nine nodes are dissatisfied and evolve into cooperators and defectors repeatedly. In other word, they are all Infectors. As a result, colored cooperators' aspiration will be higher than 3.0 as time goes so they are High-risk cooperators. Next time when one of the nodes evolves into a defector, the cooperator will be dissatisfied and evolve into a defector. Since the other nodes' aspirations are close to 4.0 and have became High-risk cooperators, chain phenomenon occurs and defectors will occupy the whole network.

When $b$ is lower, defectors' expansion requires more strict requirement. When $b = 1.2$, for the same structure shown in Fig 10, defectors can't expand. We find that before cooperators' aspirations are higher than 3.0, the nine nodes will be all satisfied in a step so that the network becomes stable with high probability. The lower $b$ makes the network stable soon if there are only nine nodes participating in the evolution. Fig 11 shows the initial structure which causes the defectors' expansion, and similar to Fig 10, sixteen nodes participate in the evolution. More Infectors make the evolutionary process last longer, so the colored nodes have enough time to increase their aspirations to higher than 3.0 and all of them will become High-risk

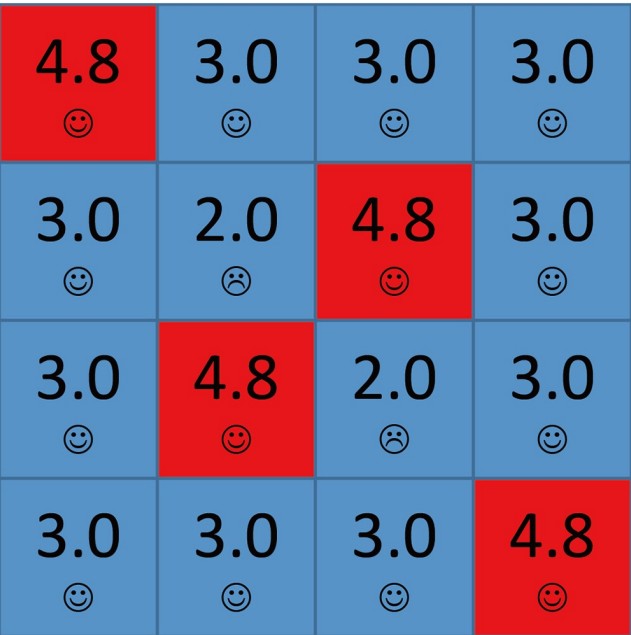

**Fig 11. The initial structure that causes defectors' expansion for $A$ = 2.4, $b$ = 1.2.** Smiling face represents satisfaction while crying face represents dissatisfaction.

cooperators. Fig 12 shows the spatial distributions of strategies and aspirations at different time steps $t$ for $A$ = 2.4, $b$ = 1.7 with the initial structure shown in Fig 10. The situation of $A$ = 2.4, $b$ = 1.2 with the initial structure shown in Fig 11 is almost the same. In fact, the above conclusion is suitable for all the $2.0 < A \leq 3.0$. Defectors' expansion requires more defectors' gathering when $b$ is lower or $A$ is higher, vice versa.

For $A > 3.0$, the nodes which have at least one $\mathcal{D}$ neighbor are dissatisfied, in other word, all the cooperators are High-risk cooperators initially. so once there are at least one defector in the network, defectors will expand to the whole network soon for any $b > 1$. The higher $b$ is, the faster defectors' expansion will be. Fig 13 shows the spatial distributions of strategies and aspirations at different time steps $t$ for $A$ = 3.2, $b$ = 1.2 with only one defector initially. Cooperators are almost impossible to survive under high aspirations.

In dynamic aspiration model, three different phases could be observed. The phase under low aspiration is similar to the fixed aspiration model because most nodes are always satisfied and their aspirations are changed in a small range. However, dynamic aspiration model plays a critical role under moderate aspiration and high aspiration, where some nodes called Infector are dissatisfied no matter they are cooperators or defectors and their strategies are changed repeatedly. As a result, their neighbors' payoff are changed and aspirations will be influenced by the evolution process, and these neighbors act as Infected nodes. Their aspirations become higher gradually but their payoffs changed repeatedly, which results in their dissatisfaction and Infected nodes will become Infectors. Once a High-risk cooperator node becomes dissatisfied, chain phenomenon happens in High-risk cooperators and defectors will expand fast.

## Conclusion

To conclude, the evolution process of the Win-Stay-Lose-Learn strategy updating rule on the prisoner's dilemma game is studied in this paper. Based on the previous work, a dynamic

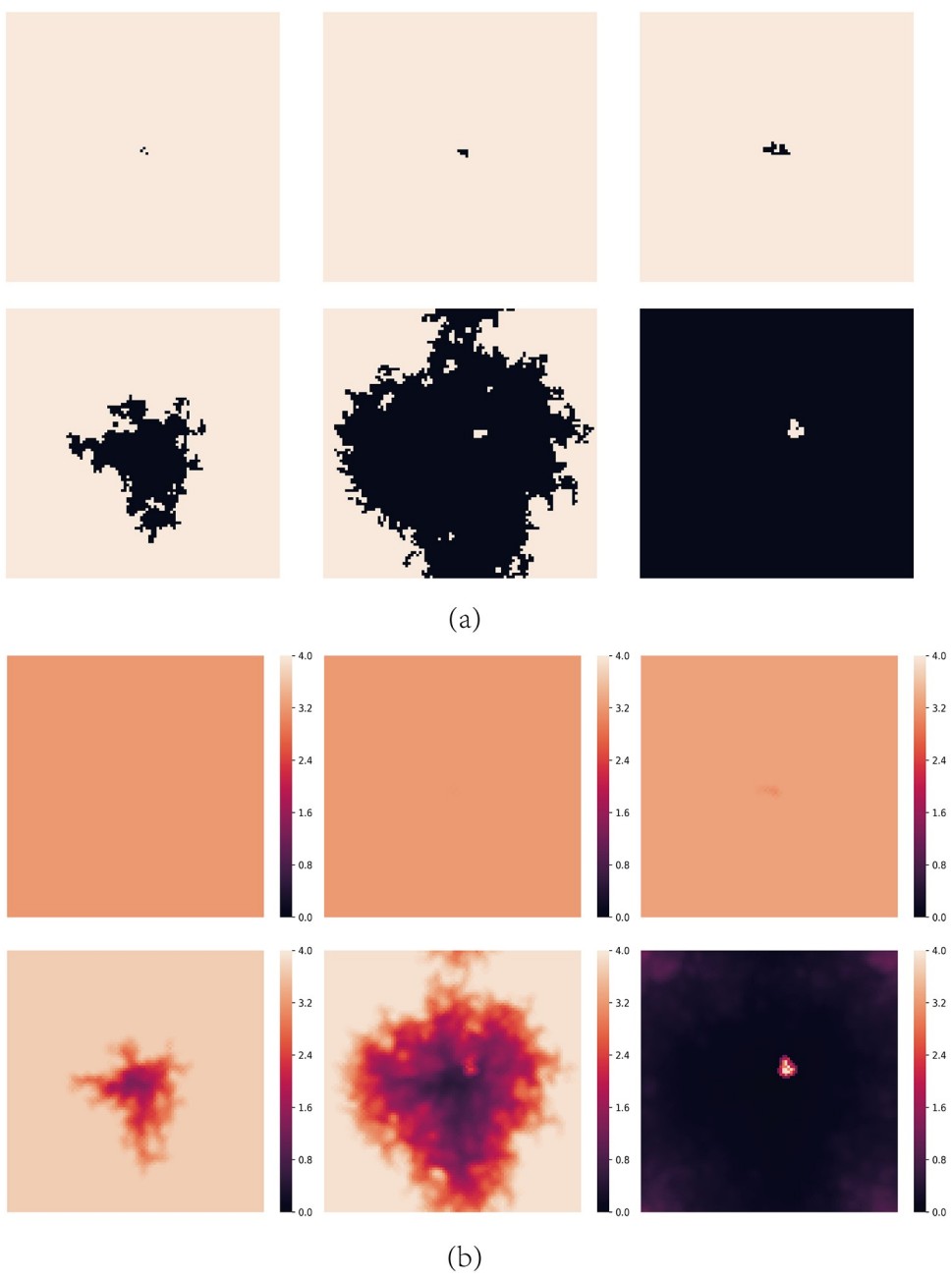

**Fig 12. Snapshots of typical distributions of strategies and aspirations at different time steps $t$ for $A = 2.4$ and $b = 1.7$ with the initial structure shown in Fig 10.** (a) represents strategies, where cooperators are depicted white and defectors are depicted black. (b) represents aspirations. The steps of them are $t = 0, 10, 100, 200, 500$ and $1000$ respectively.

aspiration model is proposed, in which players will not only change their strategies based on aspirations, but also change their aspirations due to their payoffs.

Three different phases are found. Cooperators and defectors can coexist for small values of $A$, which is called *Stable Coexistence under Low Aspiration*. Only a few cooperators will evolve into defectors then and the network will be stable immediately, which is not affected by the value of $b$. As a comparison, defectors will easily expand to the whole network for large values

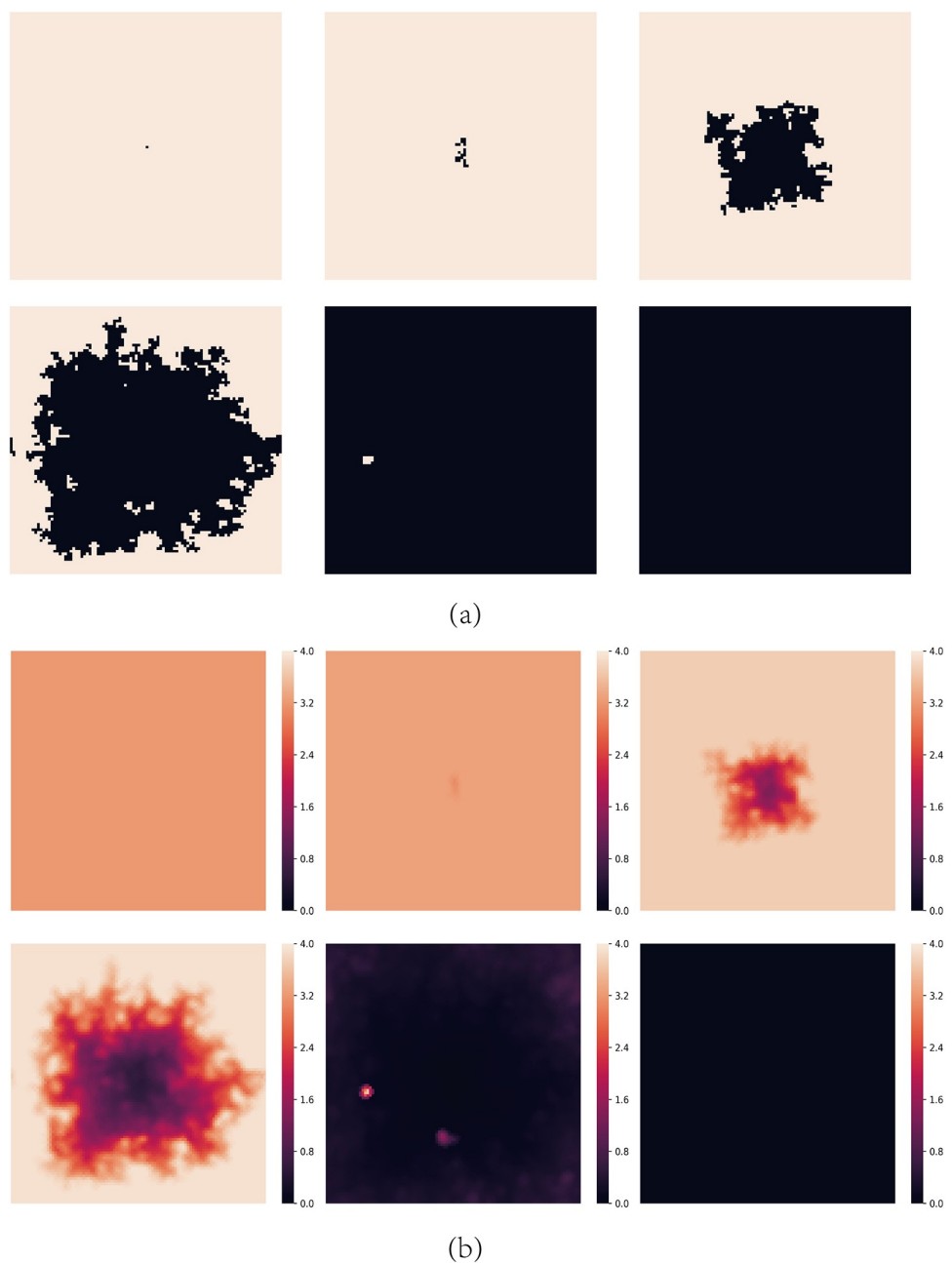

**Fig 13. Snapshots of typical distributions of strategies and aspirations at different time steps $t$ for $A = 3.2$ and $b = 1.2$ with only one defector initially.** (a) represents strategies, where cooperators are depicted white and defectors are depicted black. (b) represents aspirations. The steps of them are $t = 0, 10, 100, 200, 500$ and $1000$ respectively.

of $A$, which is called *Defection Explosion under High Aspiration* respectively. Two kinds of local structures which can lead to defectors' expansion are found, depending on the values of $b$. The most interesting phenomenon is cooperators can survive for higher $b(b \geq A)$ and die out for lower $b(b < A)$ when $1.0 < A \leq 2.0$, which is abnormal because higher $b$ should have meant that it is harder for the cooperators to survive, and it is called *Dependent Coexistence under Moderate aspiration*. The local structure leading to the defectors' expansion is that a cooperator is surrounded by one cooperator and three defectors. Dynamic aspiration plays an

important role for the above results because a constantly changing individual(Infectors) may make its neighbors' (Infected nodes) aspirations gradually rise up and they will become Infectors. At the same time, all the other cooperators' aspirations gradually rise up and they become High-risk cooperators. When a High-risk cooperator neighbors with an Infector, it will become a defector soon and chain phenomenon happens.

Our work provides a new enlightening opinion for the Win-Stay-Lose-Learn strategy updating rule. Dynamic aspiration introduces a more satisfactory explanation on population evolution laws. Under the mechanism of network reciprocity, the defectors' re-expansion is got attentions. How to avoid such unfavorable phenomenon under moderate aspirations is still a challenging problem. It is hoped that our work offers a valuable method that can help explore the principle behind prisoner's dilemma better, especially when combining with other rules which use aspiration level for personal decision making such as myopic, other-regarding preference or Pavlov-rule [52–57].

## Supporting information

**S1 File. This file (RAR format) contains the raw data used in figures with MC simulation.**
(RAR)

## Acknowledgments

We thank Marco Antonio Amaral, Xin Wang and Yuanchen Guo for discussions and suggestions.

## Author Contributions

**Conceptualization:** Zhenyu Shi, Wei Wei, Xiangnan Feng, Xing Li, Zhiming Zheng.

**Data curation:** Zhenyu Shi, Wei Wei, Xiangnan Feng, Xing Li, Zhiming Zheng.

**Formal analysis:** Zhenyu Shi, Wei Wei, Xiangnan Feng, Xing Li, Zhiming Zheng.

**Investigation:** Wei Wei, Xiangnan Feng, Xing Li, Zhiming Zheng.

**Methodology:** Zhenyu Shi, Wei Wei, Xiangnan Feng, Xing Li, Zhiming Zheng.

**Software:** Zhenyu Shi.

**Visualization:** Zhenyu Shi.

**Writing – original draft:** Zhenyu Shi, Wei Wei, Xiangnan Feng, Zhiming Zheng.

**Writing – review & editing:** Zhenyu Shi, Xing Li.

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
