## [Decision Letter · Decision Letter 0]

9 Nov 2020

PONE-D-20-31850

Dynamic aspiration based on Win-Stay-Lose-Learn rule in Spatial Prisoner's Dilemma Game

PLOS ONE

Dear Dr. Wei,

Thank you for submitting your manuscript to PLOS ONE. After careful consideration, we feel that it has merit but does not fully meet PLOS ONE’s publication criteria as it currently stands. Therefore, we invite you to submit a revised version of the manuscript that addresses the points raised during the review process.

We look forward to receiving your revised manuscript.

Kind regards,

Jun Tanimoto

Academic Editor

PLOS ONE

Journal Requirements:

2.We suggest you thoroughly copyedit your manuscript for language usage, spelling, and grammar. If you do not know anyone who can help you do this, you may wish to consider employing a professional scientific editing service.  

3.In your Data Availability statement, you have not specified where the minimal data set underlying the results described in your manuscript can be found. PLOS defines a study's minimal data set as the underlying data used to reach the conclusions drawn in the manuscript and any additional data required to replicate the reported study findings in their entirety. All PLOS journals require that the minimal data set be made fully available. For more information about our data policy, please see http://journals.plos.org/plosone/s/data-availability.

4. One of the noted authors is a group or consortium [Lorem Ipsum Consortium]. In addition to naming the author group, please list the individual authors and affiliations within this group in the acknowledgments section of your manuscript. Please also indicate clearly a lead author for this group along with a contact email address.

Reviewers' comments:

Reviewer's Responses to Questions

**Comments to the Author**

1. Is the manuscript technically sound, and do the data support the conclusions?

Reviewer #1: Yes

Reviewer #2: Partly

2. Has the statistical analysis been performed appropriately and rigorously? 

Reviewer #1: Yes

Reviewer #2: N/A

3. Have the authors made all data underlying the findings in their manuscript fully available?

Reviewer #1: Yes

Reviewer #2: Yes

4. Is the manuscript presented in an intelligible fashion and written in standard English?

Reviewer #1: Yes

Reviewer #2: No

5. Review Comments to the Author

Reviewer #1: Although what the authors did report here is simple, this work contains somehow interesting and new findings. The most important kernel in their model is dovetailing PW-Fermi with the concept of WSLS, where an agent updates his strategy by PW-Fermi (Eq. (1)) when he only obtains less than his aspiration; A_i that is individually updated by Eq. (2). This is quite simple, yet no one has ever explored. They presumed spatial game setting, and what-is-called Boundary game where T=b+1, R=1, and P=S=0.

The main result is Fig. 1, which delivers enhanced network reciprocity (cooperation fraction) when varying initial aspiration level; A_i(t=0). If the initial A_i is sufficiently less than the payoff, there is no incentive to brush-up his/ her strategy. Thus, initial cooperation fraction, perhaps 0.5, would be preserved. Interestingly, presuming an intermediate A_i, say 1.6, the network reciprocity still shows robust even in higher dilemma region (higher b). In the latter part of MS, the authors discusses why such interesting result happens.

As a whole, I feel a positive impression from the MS. Yet, I would like to give several suggestions as below so as to improve the MS more impressive to the audience of Plos One.

#1.

Although they insisted Prisoner’s Dilemma, what they explored is quite specific game, called Boundary game as I said above, which has only Chicken-type dilemma; Dg’:=(T-R)/(R-P) and zero Stag Hunt-type dilemma; Dr’=(P-S)/(R-P). Although I’ve recognized so many people favored to use Boundary game as archetype of PD under the name of ‘weak PD’, I would say this is not typical PD at all. Why people favoring is that this specific game ensures single parameter; b=T-R representing dilemma extent. What I‘ve always suggested is that one should adopt what-is-called Donor and Recipient (D & R) game where T=b, R=b-c, P=0, and S=-c meaning Dg’=Dr’=c/(b-c). It is because D & R game also allows single parameterization. More importantly, D & R game can be said more appropriate to represent PD since both Chicken- and SH-types dilemmas exist. In fact amid theoretical biologists, D & G game is much more favored as a template.

To this end, I strongly suggest the authors to get numerical results based on D & R games where varying Dg’=Dr’ ranging [0,1] instead of Boundary game. If it is the case, instead of boring line graph; Fig. 1, the authors can draw the heatmap of cooperation fraction drawn on 2D plane of Dg’=Dr’ and initial A_i, which is much more impressive.

Also the authors should explain the concept of universal dilemma strength; Dg’ and Dr’ by citing relevant literature as below for example.

Evolutionary Games with Sociophysics: Analysis of Traffic Flow and Epidemics, Springer, 2019.

Fundamentals of Evolutionary Game Theory and its Applications, Springer, 2015.

Tanimoto & Sagara; Relationship between dilemma occurrence and the existence of a weakly dominant strategy in a two-player symmetric game, BioSystems 90(1), 105-114, 2007.

Wang et al.; Universal scaling for the dilemma strength in evolutionary games, Physics of Life Reviews 14, 1-30, 2015.

Ito et al.; Scaling the phase- planes of social dilemma strengths shows game-class changes in the five rules governing the evolution of cooperation, Royal Society Open Science, 181085, 2018.

Ito et al.; Dynamic utility: the sixth reciprocity mechanism for the evolution of cooperation, Royal Society Open Science 7, 200891, 2020.

Arefin et al.; Tanimoto, J.; Social efficiency deficit deciphers social dilemmas, Scientific Reports 10, 16092, 2020.

#2.

With respect to the mechanism incurring such interesting phenomenon as I mentioned above, the authors should be take insight scope to deepen their discussion. One suggestion is that the concept of END- and EXP-periods that substantially justifies the mechanism of network reciprocity in view of time-evolution. By showing some time-evolution with coherent snapshot aside, further discussion should be explored. When referring to the concept of END- and EXP-periods, the authors should cite following relevant literatures.

Wang et al.; Insight on the so-called spatial reciprocity, Physical Review E 88, 042145, 2013.

Ogasawara et al.; Effect of a large gaming neighborhood and a strategy adaptation neighborhood for bolstering network reciprocity in a prisoner's dilemma game, Journal of Statistical Mechanics: Theory and Experiment, P12024, 2014.

Kabir et al.; Influence of bolstering network reciprocity in the evolutionary spatial Prisoner’s Dilemma game: A perspective, European Physical Journal B 91, 312, 2018.

#3.

Very recently, there have been reported some theoretical works concerning dynamic aspiration model where aspiration time-evolves like what the authors presumed. The authors should review those. For instance;

Arefin et al.; Evolution of cooperation in social dilemmas under the coexistence of aspiration and imitation mechanisms, Physical Review E 102, 032120, 2020.

Reviewer #2: The manuscript "Dynamic aspiration based on Win-Stay-Lose-Learn rule in Spatial Prisoner's Dilemma Game" begins by explaining why the evolution of cooperation is a "challenging problem", but then goes on to introduce an evolutionary model in which cooperation cannot evolve (see below). Such a model cannot teach us anything about the evolution of cooperation, and therefore has no place in a respectable journal.

The manuscript is riddled with errors. Some are mostly harmless, such as:

• Lorem Ipsum Consortium listed as an author.

• The corresponding author's mail address is given as <correspondingauthor@institute.edu>.

• Sucker's payoff being called suck's payoff.

Other errors suggest that the authors don't really know what they are talking about:

• Instead of "defection is better for selfish individuals to survive in population", the sentence should say something along the lines of "defection is favoured by evolutionary selection".

• The authors say that the evolution of defection in prisoner's dilemma "is called the tragedy of the commons". It's not! The tragedy of the commons refers to the overuse of common-pool resources by agents who act in a seemingly rational way.

• Weak prisoner's dilemma is characterised as "convenient but without losing the accuracy of the

results". The choice of one evolutionary game over the other has nothing to do with the accuracy of results.

Furthermore, the use of English is sometimes beyond interpretable. Some examples of cryptic sentences include:

• "As a result, the intrinsic structures leading to defectors’ expansion and cooperators’ survival are achieved for different evolution process, which provides a penetrating understanding of the evolution". What are 'intrinsic structures'? Which 'evolutionary processes'? What is 'penetrating understanding'?

• "However, we can easily observe numerous cooperation phenomenon in various scenarios". What are cooperation phenomena?

• "If one’s payoff is higher than its aspiration, it tends to be increased, otherwise decreased". What gets increased or decreased?

• "This is consistent with our normal perception because no one can hold a high aspiration for continued low payoff". Perception of what? Why aspiration cannot be held high in the face of continued low payoffs?

• "One can see that although cooperators take advantage in quantity in the process, defectors finally occupy the network when it is stable". How does one take advantage in quantity? And in what process?

The above issues notwithstanding, the biggest problem of this manuscript is that the model doesn't really teach us anything about the evolution of cooperation:

• When the value of the parameter A is small, every agent is satisfied, which effectively halts selection.

• When the value of the parameter A is large, every agent is dissatisfied. Selection proceeds as it normally would, causing defector domination for all but the smallest value of the temptation b.

• Finally, when the value of the parameter A is intermediate, selection can be halted for a large enough number of agents to freeze the system state before all cooperators get wiped out.

Cooperation in this model can never evolve, it can only be saved from extinction.</correspondingauthor@institute.edu>

6. PLOS authors have the option to publish the peer review history of their article (what does this mean?). If published, this will include your full peer review and any attached files.

Reviewer #1: No

Reviewer #2: No

---

## [Author Response · Author response to Decision Letter 0]

16 Dec 2020

Response to Editor: 

Thanks for your work! We have read reviewers’ advice carefully and modified the paper with response to reviewers. Besides, our response to Journal Requirements is as below:

#1：

Thank you for your advice! We have used your template for LaTeX which is downloaded in https://journals.plos.org/plosone/s/latex.

#2：

We have checked our manuscript for language usage, spelling, and grammar carefully and fixed previous errors.

#3：

All relevant data are within the manuscript and its Supporting Information files.

#4：

The author “aLorem Ipsum Consortium” is a mistake and we have deleted it.

Response to Reviewer #1: 

Thanks for your review to our work. We have read your advice carefully and modified the paper. Our response to your comments is as below:

#1.

Suggestion: Although they insisted Prisoner’s Dilemma, what they explored is quite specific game, called Boundary game as I said above, which has only Chicken-type dilemma; Dg’:=(T-R)/(R-P) and zero Stag Hunt-type dilemma; Dr’=(P-S)/(R-P). Although I’ve recognized so many people favored to use Boundary game as archetype of PD under the name of ‘weak PD’, I would say this is not typical PD at all. Why people favoring is that this specific game ensures single parameter; b=T-R representing dilemma extent. What I‘ve always suggested is that one should adopt what-is-called Donor and Recipient (D & R) game where T=b, R=b-c, P=0, and S=-c meaning Dg’=Dr’=c/(b-c). It is because D & R game also allows single parameterization. More importantly, D & R game can be said more appropriate to represent PD since both Chicken- and SH-types dilemmas exist. In fact amid theoretical biologists, D & G game is much more favored as a template.

To this end, I strongly suggest the authors to get numerical results based on D & R games where varying Dg’=Dr’ ranging [0,1] instead of Boundary game. If it is the case, instead of boring line graph; Fig. 1, the authors can draw the heatmap of cooperation fraction drawn on 2D plane of Dg’=Dr’ and initial A_i, which is much more impressive.

Also the authors should explain the concept of universal dilemma strength; Dg’ and Dr’ by citing relevant literature as below for example.

Evolutionary Games with Sociophysics: Analysis of Traffic Flow and Epidemics, Springer, 2019.

Fundamentals of Evolutionary Game Theory and its Applications, Springer, 2015.

Tanimoto & Sagara; Relationship between dilemma occurrence and the existence of a weakly dominant strategy in a two-player symmetric game, BioSystems 90(1), 105-114, 2007.

Wang et al.; Universal scaling for the dilemma strength in evolutionary games, Physics of Life Reviews 14, 1-30, 2015.

Ito et al.; Scaling the phase- planes of social dilemma strengths shows game-class changes in the five rules governing the evolution of cooperation, Royal Society Open Science, 181085, 2018.

Ito et al.; Dynamic utility: the sixth reciprocity mechanism for the evolution of cooperation, Royal Society Open Science 7, 200891, 2020.

Arefin et al.; Tanimoto, J.; Social efficiency deficit deciphers social dilemmas, Scientific Reports 10, 16092, 2020.

Response:We have read some related papers you recommended and know that many researchers were using Donor and Recipient (D & R) game to representative the PD game. However, there are also many researchers using boundary game. For instance,

Liu Y, Chen X, Zhang L, et al. Win-stay-lose-learn promotes cooperation in the spatial prisoner's dilemma game[J]. PloS one, 2012, 7(2): e30689.

Chu C, Liu J, Shen C, et al. Win-stay-lose-learn promotes cooperation in the prisoner’s dilemma game with voluntary participation[J]. Plos one, 2017, 12(2): e0171680.

Yang H X, Tian L. Enhancement of cooperation through conformity-driven reproductive ability[J]. Chaos, Solitons & Fractals, 2017, 103: 159-162.

Wu T, Wang H, Yang J, et al. The prisoner’s dilemma game on scale-free networks with heterogeneous imitation capability[J]. International Journal of Modern Physics C, 2018, 29(09): 1850077.

Our experiments show that if the sucker’s payoff S is very close to 0 (for instance, S=-0.01), the Monte Carlo simulation result is almost the same as S=0, and then we assume S=0. The boundary game is a state of PD game where S is close to P and P=b is the only variable parameter. As for Donor and Recipient game, it is a more universal model where both S and P are variable and Dg’=Dr’ can be set to allow single parameterization. Our work focuses on the phenomenon under different values of A and T(=b) and the same value of S. We have found some typical local structures which cause defectors’ expansion under S=0, where infectors and infected nodes play important roles. We also mention the two kinds of game in our revised draft and cite the references all you mentioned(in page 3 of the paper). Besides, we also use the Donor and Recipient (D & R) game to do the Monte Carlo simulation, but since different values of S may lead different results, the experimental phenomena is different from our work. I think using Donor and Recipient (D & R) game is more valuable and we will focus on it in the future.

#2.

Suggestion: With respect to the mechanism incurring such interesting phenomenon as I mentioned above, the authors should be take insight scope to deepen their discussion. One suggestion is that the concept of END- and EXP-periods that substantially justifies the mechanism of network reciprocity in view of time-evolution. By showing some time-evolution with coherent snapshot aside, further discussion should be explored. When referring to the concept of END- and EXP-periods, the authors should cite following relevant literatures.

Wang et al.; Insight on the so-called spatial reciprocity, Physical Review E 88, 042145, 2013.

Ogasawara et al.; Effect of a large gaming neighborhood and a strategy adaptation neighborhood for bolstering network reciprocity in a prisoner's dilemma game, Journal of Statistical Mechanics: Theory and Experiment, P12024, 2014.

Kabir et al.; Influence of bolstering network reciprocity in the evolutionary spatial Prisoner’s Dilemma game: A perspective, European Physical Journal B 91, 312, 2018.

Response: We have learned the concept of END- and EXP-periods from the papers you recommended and used the concept to deepen our discussion for the time-evolution of A=1.6 and b=1.2 in our revised draft (in page 5 of the paper). Besides, some new concept, including Infector, Infected node and High-risk cooperator are introduced in our revised draft (from page 6 of the paper)to explain our results better. The above concepts can help readers to understand why defectors can expand in the dynamic aspiration model together. Here are some of our modifications in the paper:

“One can see that although cooperators can survive in the END period and expand in the EXP period by forming clusters, defectors finally occupy the network when it is stable. The network reciprocity is undermined by dynamic aspirations. In dynamic aspiration models, cooperators' aspirations will become too high to endure defectors' re-expansion because of the long-term satisfaction. which is different from the fixed aspiration model.” (page 6-7 of the paper)

“In the network with random setup, during the END and EXP period, cooperators will survive and expand by the mechanism of network reciprocity. But once there is at least one Infector who has three D neighbors and one C neighbor initially, defectors will re-expand to the whole network.” (page 10 of the paper)

“Cooperators can survive by forming clusters, but since the values of b is higher, these clusters couldn't expand. The above evolution process only go through the END period and then the network have been stable.” (page 10 of the paper)

“To conclude, for moderate values of A, cooperators will survive and expand in the early stages of evolution when $b$ is lower than A, which we call END and EXP periods respectively. But according to our results, the existence of Infectors may lead to defectors' re-expansion. The core reason for this phenomenon is the cooperators increase their aspirations excessively and become the so-called High-risk cooperators, which needs to be vigilant in the evolution process of cooperation.” (page 11 of the paper)

“Dynamic aspiration plays an important role for the above results because a constantly changing individual(Infectors) may make its neighbors' (Infected nodes) aspirations gradually rise up and they will become Infectors. At the same time, all the other cooperators' aspiration are gradually rising up and they become High-risk cooperators. When a High-risk cooperator neighbors with an Infector, it will become a defector soon and chain phenomenon happens.” (page 14 of the paper)

#3.

Suggestion: Very recently, there have been reported some theoretical works concerning dynamic aspiration model where aspiration time-evolves like what the authors presumed. The authors should review those. For instance;

Arefin et al.; Evolution of cooperation in social dilemmas under the coexistence of aspiration and imitation mechanisms, Physical Review E 102, 032120, 2020.

Response: We have read the paper carefully mentioned in the comments, with some other references about dynamic aspirations. The dynamic aspiration purposed in these papers aim the coexistence of aspiration and imitation mechanisms or how cooperation spreads, which is related but has different concentrations with our work. Our work’s focus is how defectors can expand with the impact of Infectors and High-risk cooperators in dynamic aspiration model. Besides, we have cited these papers you mentioned in our revised draft(in page 2 of the paper).

Response to Reviewer #2: 

Thanks for your review to our work. We have read your advice carefully and modified the paper. Our response to your comments is as below:

In your review comments, you mentioned some language errors in our paper so that we have checked our paper and fixed them:

(1)

Suggestion: Lorem Ipsum Consortium listed as an author.

Response: “Lorem Ipsum Consortium” has been deleted. (page 1 of the paper)

(2)

Suggestion: The corresponding author's mail address is given as .

Response: The corresponding author's mail address has been corrected.(page 1 of the paper)

(3)

Suggestion: Sucker's payoff being called suck's payoff.

Response: “Suck's payoff” has been corrected as “sucker's payoff”. (page 3 of the paper)

(4)

Suggestion: Instead of "defection is better for selfish individuals to survive in population", the sentence should say something along the lines of "defection is favoured by evolutionary selection".

Response: The sentence "defection is better for selfish individuals to survive in population" is not very appropriate and it has been corrected as "defection is favoured by evolutionary selection" by your suggestion. (page 1 of the paper)

(5)

Suggestion: The authors say that the evolution of defection in prisoner's dilemma "is called the tragedy of the commons". It's not! The tragedy of the commons refers to the overuse of common-pool resources by agents who act in a seemingly rational way.

Response: We are sorry that we have not figured out the meaning of “tragedy of the commons" and used it improperly, and "which is called the tragedy of the commons" has been deleted. (page 1 of the paper)

(6)

Suggestion: Weak prisoner's dilemma is characterised as "convenient but without losing the accuracy of the results". The choice of one evolutionary game over the other has nothing to do with the accuracy of results.

Response: The statement "convenient but without losing the accuracy of the results" is inaccurate. We have rewritten part 1 of the model section as:

“Without a loss of generality, we set R = 1 and P = 0. And to ensure single parameter, there are some typical representative sub-classes of PD game, e.g., Donor & Recipient (D & R) game which assumes T + R = 1 [43–49] and boundary game which assumes T = b and S = 0 [16, 33]. In this paper, we use boundary game because what we mainly study is the impact of T on evolution process. It should be noted that in PD game, S < 0. However, Our experiment shows that if S is close to 0(for instance, S = −0.01), then the Monte Carlo simulation result is almost the same as S = 0, that’s why boundary game can assume S = 0.” (page 3 of the paper)

(7)

Suggestion: "As a result, the intrinsic structures leading to defectors’ expansion and cooperators’ survival are achieved for different evolution process, which provides a penetrating understanding of the evolution". What are 'intrinsic structures'? Which 'evolutionary processes'? What is 'penetrating understanding'?

Response: The expression “intrinsic structures” , “evolutionary processes” and “penetrating understanding” are inaccurate. We have rewritten the last few sentences in the abstract section as:

“Furthermore, a deep analysis is performed on the local structures which cause defector's re-expansion, and the concept of END- and EXP-periods are used to justify the mechanism of network reciprocity in view of time-evolution. As a result, we find some typical feature nodes which we called Infectors, Infected nodes and High-risk cooperators respectively. Compared to fixed aspiration model, dynamic aspiration introduces a more satisfactory explanation on population evolution laws and can promote deeper comprehension for the principle of prisoner's dilemma.”(page 1 of the paper)

(8)

Suggestion: "However, we can easily observe numerous cooperation phenomenon in various scenarios". What are cooperation phenomena?

Response: The sentence "However, we can easily observe numerous cooperation phenomenon in various scenarios" should have some references. “Cooperation phenomenon” in this sentence is a concept from other papers and we have cited it and the complete sentence has been revised as “However, we can easily observe numerous cooperation phenomenon in various scenarios, e.g., animals will cooperate to obtain food instead of preying alone [7]; companies will set appropriate commodity prices instead of maliciously cutting prices [8]; humans will choose to obey the order instead of jumping in line, etc [9].”(page 1 of the paper)

(9)

Suggestion: "If one’s payoff is higher than its aspiration, it tends to be increased, otherwise decreased". What gets increased or decreased?

Response: The sentence "If one’s payoff is higher than its aspiration, it tends to be increased, otherwise decreased." is not clear and we have revised as “If one's payoff is higher than its aspiration, the aspiration tends to be increased, otherwise decreased.”(page 2 of the paper)

(10)

Suggestion: "This is consistent with our normal perception because no one can hold a high aspiration for continued low payoff". Perception of what? Why aspiration cannot be held high in the face of continued low payoffs?

Response: This sentence "This is consistent with our normal perception because no one can hold a high aspiration for continued low payoff" doesn’t express accurately enough. We have combined it with the new references we cited as”Some researchers also paid attention to this and had some related researches[37–39]. Our view is consistent with the view in [38] that in the middle of a crisis, individuals tend to lower what their aspirations in interactions, vice versa.” (page 3 of the paper)

(11)

Suggestion: "One can see that although cooperators take advantage in quantity in the process, defectors finally occupy the network when it is stable". How does one take advantage in quantity? And in what process?

Response: The sentence "One can see that although cooperators take advantage in quantity in the process, defectors finally occupy the network when it is stable" has unclear meaning and we have revised it as “One can see that although cooperators can survive in the END period and expand in the EXP period, defectors finally occupy the network when it is stable.” (page 6 of the paper)

(12)

Suggestion: The above issues notwithstanding, the biggest problem of this manuscript is that the model doesn't really teach us anything about the evolution of cooperation:

• When the value of the parameter A is small, every agent is satisfied, which effectively halts selection.

• When the value of the parameter A is large, every agent is dissatisfied. Selection proceeds as it normally would, causing defector domination for all but the smallest value of the temptation b.

• Finally, when the value of the parameter A is intermediate, selection can be halted for a large enough number of agents to freeze the system state before all cooperators get wiped out.

Cooperation in this model can never evolve, it can only be saved from extinction.

Response: Thank you for this detailed suggestion to our work. Your main opinion is that cooperation’s evolution is the most important. In our manuscript, we analyze the social dilemma from another angle. In most previous studies, the re-expansion of the defectors has not been noticed. This paper finds the key local structure that influences the re-expansion of the defectors through related research. I think this is as important as the evolution of cooperation. If such local structures can be noticed and avoided, cooperation can expand and survive in a better environment. Besides, our next step is to focus on how to avoid related structures and their negative impact to promote cooperation. We also make much modification in the paper to deepen our discussion of the relationship between the cooperation’s evolution and our results. Such as:

“To conclude, for small values of A, initial cooperators could coexist with defectors and neither of them could expand. Low aspiration means both strategies and aspirations are long-term stable.” (page 5 of the paper)

“One can see that although cooperators can survive in the END period and expand in the EXP period by forming clusters, defectors finally occupy the network when it is stable.The network reciprocity is undermined by dynamic aspirations. In dynamic aspiration models, cooperators' aspirations will become too high to endure defectors' re-expansion because of the long-term satisfaction. which is different from the fixed aspiration model.” (page 6-7 of the paper)

“In other words, defectors could expand and occupy the network finally even if they are very few initially. This result is different from the result of fixed aspiration model and it may be caused by some special local structures which could lead to the too high aspirations of cooperators.” (page 7 of the paper)

“In the network with random setup, during the END and EXP period, cooperators will survive and expand by the mechanism of network reciprocity. But once there is at least one Infector who has three D neighbors and one C neighbor initially, defectors will re-expand to the whole network.” (page 10 of the paper)

“Cooperators can survive by forming clusters, but since the values of b is higher, these clusters couldn't expand. The above evolution process only go through the END period and then the network have been stable.” (page 10 of the paper)

“To conclude, for moderate values of A, cooperators will survive and expand in the early stages of evolution when $b$ is lower than A, which we call END and EXP periods respectively. But according to our results, the existence of Infectors may lead to defectors' re-expansion. The core reason for this phenomenon is the cooperators increase their aspirations excessively and become the so-called High-risk cooperators, which needs to be vigilant in the evolution process of cooperation.” (page 11 of the paper)

“Dynamic aspiration plays an important role for the above results because a constantly changing individual(Infectors) may make its neighbors' (Infected nodes) aspirations gradually rise up and they will become Infectors. At the same time, all the other cooperators' aspiration are gradually rising up and they become High-risk cooperators. When a High-risk cooperator neighbors with an Infector, it will become a defector soon and chain phenomenon happens.” (page 14 of the paper)

“Under the mechanism of network reciprocity, the defectors' re-expansion is got attentions. How to avoid such unfavorable phenomenon under moderate aspirations is still a challenging problem.” (page 14 of the paper)

---

## [Decision Letter · Decision Letter 1]

17 Dec 2020

Dynamic aspiration based on Win-Stay-Lose-Learn rule in Spatial Prisoner's Dilemma Game

PONE-D-20-31850R1

Dear Dr. Wei,

We’re pleased to inform you that your manuscript has been judged scientifically suitable for publication and will be formally accepted for publication once it meets all outstanding technical requirements.

Kind regards,

Jun Tanimoto

Academic Editor

PLOS ONE

Additional Editor Comments (optional):

Reviewers' comments:

Reviewer's Responses to Questions

**Comments to the Author**

1. If the authors have adequately addressed your comments raised in a previous round of review and you feel that this manuscript is now acceptable for publication, you may indicate that here to bypass the “Comments to the Author” section, enter your conflict of interest statement in the “Confidential to Editor” section, and submit your "Accept" recommendation.

Reviewer #1: All comments have been addressed

2. Is the manuscript technically sound, and do the data support the conclusions?

Reviewer #1: Yes

3. Has the statistical analysis been performed appropriately and rigorously? 

Reviewer #1: Yes

4. Have the authors made all data underlying the findings in their manuscript fully available?

Reviewer #1: Yes

5. Is the manuscript presented in an intelligible fashion and written in standard English?

Reviewer #1: Yes

6. Review Comments to the Author

Reviewer #1: The revise MS seems sufficiently persuasive to all suggestions I gave. Hence, I can nod that this MS can be welcomed to the journal.

7. PLOS authors have the option to publish the peer review history of their article (what does this mean?). If published, this will include your full peer review and any attached files.

Reviewer #1: No

---

## [Editor Report · Acceptance letter]

23 Dec 2020

PONE-D-20-31850R1

Dynamic aspiration based on Win-Stay-Lose-Learn rule in Spatial Prisoner's Dilemma Game

Dear Dr. Wei:

I'm pleased to inform you that your manuscript has been deemed suitable for publication in PLOS ONE. Congratulations! Your manuscript is now with our production department.

Kind regards,

on behalf of

Prof. Jun Tanimoto

Academic Editor

PLOS ONE